# Untargeted lipidomics reveals unique lipid signatures of extracellular vesicles from porcine colostrum and milk

Rafaela Furioso Ferreira[1,2]*, Morteza H. Ghaffari[1], Fabrizio Ceciliani[3], Manuela Fontana[4], Donatella Caruso[4,5], Matteo Audano[4], Giovanni Savoini[5], Alessandro Agazzi[5], Vladimir Mrljak[2], Helga Sauerwein[1]

1 Institute of Animal Science, Physiology Unit, University of Bonn, Bonn, Germany, 2 Faculty of Veterinary Medicine, University of Zagreb, Zagreb, Croatia, 3 Department of Veterinary Medicine and Animal Sciences, Università degli Studi di Milano, Lodi, Italy, 4 Unitech OMICs, Università degli Studi di Milano, Milano, Italy, 5 Department of Pharmacological and Biomolecular Sciences, Università degli Studi di Milano, Milano, Italy

* rafaelaff.vet@gmail.com

## Abstract

Extracellular vesicles (EV) are membranous vesicles considered as significant players in cell-to-cell communication. Milk provides adequate nutrition, transfers immunity, and promotes neonatal development, and milk EV are suggested to play a crucial role in these processes. Milk samples were obtained on days 0, 7, and 14 after parturition from sows receiving either a standard diet (ω-6:ω-3 = 13:1) or a test diet enriched in ω-3 (ω-6:ω-3 = 4:1). EV were isolated using ultracentrifugation coupled with size exclusion chromatography, and characterized by nanoparticle tracking analysis, transmission electron microscopy, and assessment of EV markers via Western blotting. The lipidome was determined following a liquid chromatography–quadrupole time-of-flight mass spectrometry approach. Here, we show that different stages of lactation (colostrum vs mature milk) have a distinct extracellular vesicle lipidomic profile. The distinct lipid content can be further explored to understand and regulate milk EV functionalities and primordial for enabling their diagnostic and therapeutic potential.

## Introduction

Milk is a nutrient-rich biofluid with a complex composition, including abundant extracellular vesicles (EV), as reported for different species [1–5]. The EV are membrane bound nano-sized vesicles, mediators of intercellular communication in short and longer-range signaling events for their ability to transfer proteins, lipids, genetic material, and other metabolites between different cell types [6–8]. EV cargo and composition change and adapt to external and local stimuli [9]; sorting specific compounds during the EV biogenesis may not always mirror the protein composition of the parental cells [10].

Milk contains various particles aside than EV, including milk fat globules (MFG) and casein micelles, which present an additional challenge for EV isolation compared to other biological

**Data Availability Statement:** We have added a clear description of how interested readers may access the raw data on the repository of the University of Milan for ease of access. This

information has been provided as part of the Data Availability Statement and includes the specific DOI links: For positive ion mode: https://doi.org/10.13130/RD_UNIMI/UZI5O8 For negative ion mode: https://doi.org/10.13130/RD_UNIMI/0HA5IL.

**Funding:** This work was supported with the European Union's Horizon 2020 research and innovation programme H2020-MSCA- ITN-2017-EJD: Marie Skłodowska-Curie Innovative Training Networks (European Joint Doctorate) [Grant agreement nº: 765423, 2017] – MANNA."

**Competing interests:** The authors have declared that no competing interests exist.

**Abbreviations:** Cer, Ceramides; Chol, cholesterol; ESCRT, Endosomal Sorting Complex Required for Transport; EV, Extracellular Vesicles; GL, Group L (linseed oil); GS, Group S (soybean oil); ILV, Intraluminal Vesicles; MGF, Milk Fat Globules; MVE, Multivesicular Endosomes; NTA, Nanoparticle Tracking Analysis; PC, Phosphatidylcholine; QTOF, Quadrupole Time-of-Flight; SM, Sphingomyelin; TEM, Transmission Electron Microscopy; TG, Triacylgycerols.

fluids due to potential overlap in size and structural components [11]. While milk EV are characterized by a phospholipid bilayer and originate from multiple cellular sources [12], MFG are primarily composed of triacylglycerols and emerge from the endoplasmic reticulum of mammary gland alveolar epithelial cells, until released as lipid droplets encased in a phospholipid trilayer [13, 14]. Given the intricate composition of milk, careful consideration is necessary when selecting the methodology for EV isolation [15].

Milk EV act on the development of the infant's intestinal tract [16, 17] and immune system [18, 19]. Recently described functions also include the capability to induce cell proliferation [20] and enhancing the epithelial barrier [21, 22]. Milk EV protect their content from enzymatic degradation [23] and can remain bioactive even after pasteurization [24], while their content can be up-taken by various cell types, including cross-species [25, 26]. While the study of milk EV is important given their biological function and the use of milk and dairy products as food [27], milk EV have also been targeted as a potential natural source of lipid-based nanocarriers for drug delivery systems [28–30].

Nanoparticle-delivery systems are a relatively new and rapidly developing technology that allows the delivery of unstable molecules, enhances the efficacy of therapeutic agents, and may allow site-specific, target-oriented delivery agents [31, 32]. Despite many advantages, various factors have hindered the clinical use of drug-delivery nanoparticles, such as the high cost, poor stability, non-specific binding, toxicity issues, and large-scale production [33]. Milk EV could potentially overcome some of the limitations of synthetic liposomes due to their superior half-life in circulation and their bioavailability, avoidance of degradation, and cross-species tolerance [34–36]. Therefore, understanding the structural, functional, and stability characteristics of milk EV is highly sought.

Lipids comprise the main structural fraction of the EV membrane and are essential for EV formation, release, targeting, and uptake [37, 38]. EV protein and nucleic acid (mRNA and microRNA) content has been extensively characterized, whereas their lipid components have been largely overlooked, including in milk EV [39]. Sphingomyelin, phospholipids, and cholesterol are lipid classes commonly found in cell membranes and, therefore, in EV [40]. However, their relative abundance and enrichment of other lipid classes may depend on the producing cell type [41] and the cell's physiological state [42, 43].

The ratio of dietary ω-6 to ω-3 polyunsaturated fatty acids (PUFA) is relevant both in human and livestock nutrition [44], as the use of diets enriched in ω-3 PUFA has been associated with the prevention of many chronic diseases [45, 46]. In pig production, diets with a lower ω-6 to ω-3 ratio were shown to exert anti-inflammatory and antioxidative properties [47, 48]. Recent works suggest that the metabolism of the sows and their piglet's health might also be modulated [49, 50]. The dietary ω-6:ω-3 ratio is also known to affect the fatty acid composition in milk [51, 52], plasma, serum, and ileal proteomes proteome [53, 54] and muscle metabolism [55, 56]. Still, the lipid composition of milk EV, which can be affected by the maternal diet, is largely unexplored. Furthermore, pigs are also a valuable model for translational research due to their similarity with humans in terms of physiology and metabolism, the comparable nutritional requirements, and similar developmental patterns [57, 58]. Together with similar physiological reactions to xenobiotically induced changes, pre-clinical toxicological trials in pigs for novel drug delivery systems are feasible [59].

This study provides new insights into the lipidomic characterization of porcine milk EV, filling a significant gap in our understanding of how maternal nutrition influences the lipid composition of these vesicles. We hypothesized that the intake of two different diets, grain silage and grass legumes, characterized by a different ratio of ω-6:ω-3 fatty acids, would significantly alter the lipid composition of milk vesicles. The objectives of this study were to provide

a comprehensive lipidomic characterization of porcine milk EV derived from sow milk in different stages of lactation and to assess how different ratios of ω-6:ω-3 fatty acids in the gestation and lactation diets of sows affect the lipidomic composition of milk EV.

## Material and methods

### Note on the terminology

The use of nomenclature used in this manuscript follows the MISEV2023 guidelines (https://www.isev.org/misev), endorsing that "extracellular vesicle" remains the general term for particles released from cells wrapped by a lipid bilayer and without a functional nucleus, and exosomes of the endosomal-origin EV subtype [60].

### Animals and sample collection

Sixteen multiparous sows with similar body weight and body condition scores were randomly allocated to one of two dietary treatments: the control group (Group S—GS) received a standard diet with a ω-6:ω-3 ratio of 13:1 from day 28 of gestation until farrowing and 10:1 during lactation. The treatment group (Group L—GL) was fed a diet with a low ω6:ω3 ratio (4:1 from day 28 of gestation until the end of lactation, i.e., 24 days after parturition). The ω-6 and ω-3 fatty acids for this study were derived from soybean oil (GS) and linseed oil (GL), respectively. The complete information on the composition of the basal diets of the sows and the adjustments to the final ratios of ω-6:ω-3 PUFAs are provided in S1 Table. All sows in this study were at second parity, with the insemination for second parity performed at 13 months of age and the average age at parturition being around 17 months. Details on body weight and body condition of sows from this study during gestation and at the end of lactation are provided in S1 Table. All diets were calculated to be isonitrogenous and isoenergetic and to meet or exceed the estimated nutrient requirements for sows during gestation and lactation [61]. For the present study, we have randomly selected 10 sows (n = 5 from each group) from the experiment to collect samples during the natural milk ejection at day 0 (colostrum), 7, and 14 (mature milk) postpartum, frozen immediately after collection, and kept at −80 ˚C until EV isolation.

The animal experiment was conducted on a commercial swine farm (Arioli and Sangalli Agricultural Company S. S.; Genzone, Italy), and the effects of a low versus a high ω-6:ω-3 ratio on performance, colostrum and milk fatty acid profiles have already been published [47]. The experimental procedures performed in this study followed the European Union Guidelines concerning the protection of experimental animals, with approval by the local authority for animal welfare affairs (the Ethical Committee of the University of Milan [OPBA 67/2018] and the Italian Ministry of Health [authorization n. 168/2019 PR]).

### Isolation of EV from colostrum and milk

Milk EV were isolated using ultracentrifugation coupled with size exclusion chromatography to enrich milk-derived small EV ($< 200$ nm) as previously described [62, 63]. Samples with an initial volume of 2 mL were centrifuged at $4,000 \times g$ for 30 min at 4 ˚C. Avoiding the fat layer, approximately 1.5 mL of skimmed milk were removed and centrifuged at 12,000 x $g$ for 30 min at 4 ˚C. The supernatant was centrifuged at 100,000 x $g$ for 1 h at 4 ˚C. The EV pellet formed on top of the firm casein pellet was carefully removed and resuspended in PBS. After further centrifugation of $150,000 \times g$ for 2 h at 4 ˚C, the EV pellet was collected, removed from the firm casein pellet, and resuspended in PBS to 500 μL. This suspension was then loaded on a qEVoriginal 35 nm size exclusion chromatography column (Izon Science, Oxford, UK),

following the manufacturer's instructions. After the void volume, 4 fractions of each 500 μL were collected. Fractions 2 and 3 contained a higher concentration of small EV and were pooled for the following analysis. Total protein concentration was immediately estimated using Nanodrop ND-1000 Spectrophotometer (Thermo Fisher Scientific, Wilmington, USA). The pooled Fractions 2 and 3 containing the small EV isolation was separated in different aliquots for the distinct further analysis in order to avoid freeze-thawing cycles. The isolates were stored at -80 ˚C until further analysis. This temperature was selected explicitly for its acknowledged ability to maintain the structural and functional integrity of EVs over extended storage durations, as previously reported [29, 64–66].

**Nanoparticle Tracking Analysis (NTA).**   The concentration and size distribution of particles in the fractions were measured with NanoSight NS300 (NTA 3.1, Malvern Panalytical, Malvern, UK). Fractions were diluted to 1:25–1:1,000-fold in PBS to keep the number of particles in the field between 50 and 200/frame. Three videos of 30-sec duration were captured with camera gain set at 5, camera level set at 12, and detection threshold set at 5.

**Transmission Electron Microscopy (TEM).**   For electron microscopy analysis, 20 μL of purified EV samples were deposited on Parafilm (Bemis, Neenah, WI, USA). A 400-mesh copper grid (45 μm square, Agar Scientific, Essex, UK) coated with 1% Piolofom (Plano GmbH, Wetzlar, Germany) was immediately placed onto a drop surface facing the filmed side. Grids were incubated for 5 min at RT and then carefully drained on low-lint wipes by vertically touching the tissue, followed by a 5 min incubation at RT with a large drop of distilled water containing Bacitracin as wetting agent (1 mg/100 mL). Grids were drained after 5 min and negatively stained with 1% uranyl acetate (w/w) (SERVA, Rosenheim, Germany) with the filmed side downwards and drained again on low-lint wipes with the filmed side upwards. Grids were transferred onto a graded and numbered rubber plate in a glass petri dish and given time to dry completely. Samples were then observed with a Zeiss 109 T transmission electron microscope (Carl Zeiss QEC GmbH, Köln, Germany) upgraded with a Point Electronics update kit. Images were taken with a TRS wide-angle dual-speed 2K CCD camera with a YAG scintillator.

**Verification of EV marker by Western Blotting.**   Proteins from all fractions were dissolved in a reducing sample buffer, boiled, and loaded on 12% SDS-PAGE. The fractionated proteins were transferred to a polyvinylidene difluoride membrane using the Trans-Blot Turbo transfer unit for 30 min at 1 A and 25 V (Bio-Rad Laboratories, Munich, Germany). The membranes were blocked with Tris-buffered saline containing 0.05% Tween 20 (TBST) and Rotiblock (Carl Roth, Karlsruhe, Germany) for 60 min at RT. Blots were incubated with mouse monoclonal anti-TSG101 IgG2a-κ (sc-7964, Santa Cruz Biotechnology, Dallas, TX, USA) overnight at 4 ˚C, washed three times with TBS-T, and incubated for 120 min at RT with secondary mouse IgGκ light chain binding protein (m-IgGκ BP) conjugated to horseradish peroxidase (sc-516102, Santa Cruz Biotechnology). After washing, the immune complex was detected with Amersham ECL Select Western Blotting Detection Reagent chemiluminescence detection system (RPN2235, Cytiva, Freiburg, Germany). Imaging was performed with a VersaDoc MP4000 imaging system (Bio-Rad, Hercules, CA, USA).

**Lipid extraction.**   Milk EV samples were lipid extracted as previously described [65] with minor modifications. Briefly, two aliquots (200 μL) from each sample were added to internal standards (Splash Lipidomix Internal Standards Avanti Polar and a mix of $^{13}$C-Palmitic and $^{13}$C-Linoleic Acids) and extracted according to the Folch method [67]. The organic residue was reconstituted with 2-propanol: acetonitrile (90:10, vol/vol), 0.1% formic acid, and 10 mM ammonium acetate. Each sample was extracted in duplicate, and 2 runs were performed for each extraction.

## Lipidomics analysis

Samples were analyzed at the UNITECH platform OMICS (Università degli Studi di Milano, Italy) as follows: 2 μL of the sample, for both positive and negative ion modes, were separated by liquid chromatography with a Kinetex EVO C18 column (2.1 × 100 mm, 1.7 μm; Phenomenex) at 45˚C, connected to an ExionLC AD system (ABSciex) maintained at 15˚C. Separated metabolites were then ionized through an electrospray ionization source and analyzed in a TripleTOF 6600 (quadrupole time-of-flight, QTOF, ABSciex) mass spectrometer. Mobile phases were (A) water with 0.1% formic acid and 10 mM ammonium acetate/acetonitrile (60:40) and (B) 2-propanol with 0.1% formic acid and 10 mM ammonium acetate/acetonitrile (90:10). The following elution gradient was used: 0 min 45% B; 2 min 45% B; 12 min 3% B; 17 min 3% B; 17.10 min 45% B; 20 min 45% B. The flow rate was 0.4 mL/min. Electrospray ionization and mass spectrometer conditions were set as presented in Table 1.

Quality control (QC) samples were analysed every 5 biological samples to assess the variance observed in the data throughout the sample preparation, data acquisition and data preprocessing steps. The QC samples are qualitatively and quantitatively representatives of the entire list of samples included in the study, providing an average of all the metabolomes analysed. S1 Fig. exhibits 4 QC representative ion chromatograms, in positive and negative ionization (two each). The raw data supporting the findings of this study are available on the University of Milan repository accessible at https://doi.org/10.13130/RD_UNIMI/UZI5O8 for positive ion mode and https://doi.org/10.13130/RD_UNIMI/0HA5IL for negative ion mode.

**Data processing.** Data processing was carried out using the untargeted data processing program MSDIAL (v. 4.24; http://prime.psc.riken.jp/compms/msdial/main.html) with LipidBlast database v. 68 (https://fiehnlab.ucdavis.edu/projects/LipidBlast). This database contains 81 lipid classes, 377,313 molecules, and 554,041 spectra in positive polarity, and 94 lipid classes, 356,477 molecules, and 792,757 spectra in negative polarity. Data are expressed as the ratio of analyte to the internal standard area and total protein content.

**Lipid nomenclature.** Lipid classification in categories, main class, and subclass was carried out according to the LIPID MAPS database [68]. Lipids are abbreviated as follows: fatty acids–acylcarnitines (CAR), free fatty acids (FA), n-acyl ethanolamines (NAE), n-acyl glycines (NAGly), n-acyl glycyl serines (NAGlySer), n-acyl ornithines (NAOrn), oxidized fatty acids (OxFA); glycerolipids–acyl diacylglyceryl glucuronides (ADGGA), diacylglycerols (DG), diacylglyceryl glucuronides (DGGA), diacylglyceryl trimethylhomoserine/diacylglyceryl

**Table 1. ESI and mass spectrometer parameters.**

| Item | Lipidomic IDA POS | Lipidomic IDA NEG |
|---|---|---|
| Ionization | POS | NEG |
| Source temperature | 350 ˚C | 350 ˚C |
| Curtain Gas (CUR) | 35 | 35 |
| GS 1 | 55 | 55 |
| GS 2 | 65 | 65 |
| Ion Spray Voltage | 5500 V | -4500 V |
| Declustering Potential (DP) | 50 V | -50 V |
| Collision Energy | 35V | -40V |
| Collision Energy Spread | 15 | 20 |
| TOF MS Mass Range | 140–2000 Da | 150–1100 Da |
| IDA acquisition Mass Range | 50–2000 Da | 50–1100 Da |
| Top N | 18 | 18 |

hydroxymethyl-n,n,n-trimethyl-β-alanines (DGTS), diacylglyceryl-3-o-carboxyhydroxy-methylcholines (DGCC), digalactosyldiacylglycerols (DGDG), lysodiacylglyceryl trimethylho-moserine/lysodiacylglyceryl hydroxymethyl-n,n,n-trimethyl-β-alanines (LDGTS), monoacylglycerols (MG), triacylglycerols (TG); glycerophospholipids–bismonoacylglycero-phosphates (BMP), cardiolipins (CL), dimethyl-phosphatidylethanolamines (DMPE), hemi-bismonoacylglycerophosphates (HBMP), lysophophatidylcholines (LPC), lysophosphatidic acids (LPA), lysophosphatidylethanolamine (LPE), n-acyl-lysophosphatidylserines (LNAPS), n-monomethyl phosphatidylethanolamines (MMPE), phosphatidic acids (PA), phosphatidyl-cholines (PC), phosphatidylethanol (PetOH), phosphatidylethanolamines (PE), phosphatidyl-glycerols (PG), phosphatidylinositols (PI), phosphatidylmethanol (PMeOH), phosphatidylserines (PS); prenol lipids—coenzyme Q (CoQ10); sphingolipids–acylhexosylcer-amides (AHexCer), ceramide 1-phosphates (CerP), ceramide phosphoethanolamines (PE-Cer), ceramide phosphoinositol (PI-Cer), ceramides (Cer), hexosylceramides (HexCer), sphingomyelins (SM), sulfatides (SHexCer), sulfonolipid (SL); and sterol lipids—cholesteryl ester (CE) sterol esters (SE), sterols (ST). Alkyl ether and plasmalogen linkages are denoted by O- and P- respectively. The side-chain structures are denoted as carbon chain length:number of double bonds provided for each chain where they could be determined, or as the total number of all carbons and double bonds where individual chains could not be determined. To reduce process-based sources of variation and extreme biological outliers, QC samples were injected every 5 biological samples.

**Univariate and multivariable statistical analysis.** MS data were checked for integrity, and variables containing more than 20% missing values (i.e., values lower than the detection limit) were not considered for the statistical analysis using the MetaboAnalyst 5.0 webtool [68]. When present, missing values were imputed by Bayesian principal component analysis by MetaboAnalyst 5.0. The data were then transformed by log10-transformation and Pareto scaled to correct for heteroscedasticity, reduce the skewness of the data, and reduce mask effects [52]. Volcano plots were created to provide an overview of the lipid profiles in two diets (GS vs. GL) and different sampling times (day 0 vs. 7; day 0 vs 14; day 7 vs.14) using the log-transformed data. Volcano plots relate fold change (FC) to statistical significance. Significance was set using a t-test FDR adjusted p-value threshold at 0.05 and fold change threshold at 2. The heatmaps were clustered by Euclidean distance and Ward's minimum variance method (ward.D).

## Results

### Characterization of the milk and colostrum EV isolation

In the first step of the study, sow milk small EVs were purified to homogeneity, and the porcine milk small EV were characterized by NTA, TEM, and Western blotting. The ANOVA mixed model using the NTA measurements revealed that the treatments and the interaction between time and treatment yielded no differences in EV size and concentration. At the same time, the timepoints alone (colostrum vs milk) did (Fig 1A). The average EV diameter (nm) did not differ between the two groups (GL = 162 ± 14.7; GS = 156 ± 20.6; means ± SD), and the EV population was slightly smaller in colostrum (147 ± 21.3) than in mature milk (165 ± 14.2 at day 7 and 166 ± 11 at day 14). The average concentration was also similar in both groups, with a higher concentration of EV in colostrum ($5.65 \times 10^{11} \pm 2.7 \times 10^{11}$ particles/mL) than in mature milk ($2.48 \times 10^{11} \pm 9.7 \times 10^{10}$ at day 7 and ($1.98 \times 10^{11} \pm 8.64 \times 10^{10}$ at day 14—Fig 1B). To confirm the successful isolation of small EV, we demonstrated the EV marker protein TSG101 in both groups by Western Blotting (Fig 1D). The expression of other top EV markers, such as CD9, CD63, CD81, Heat Shock Protein Family A, and flotillin 1, have also

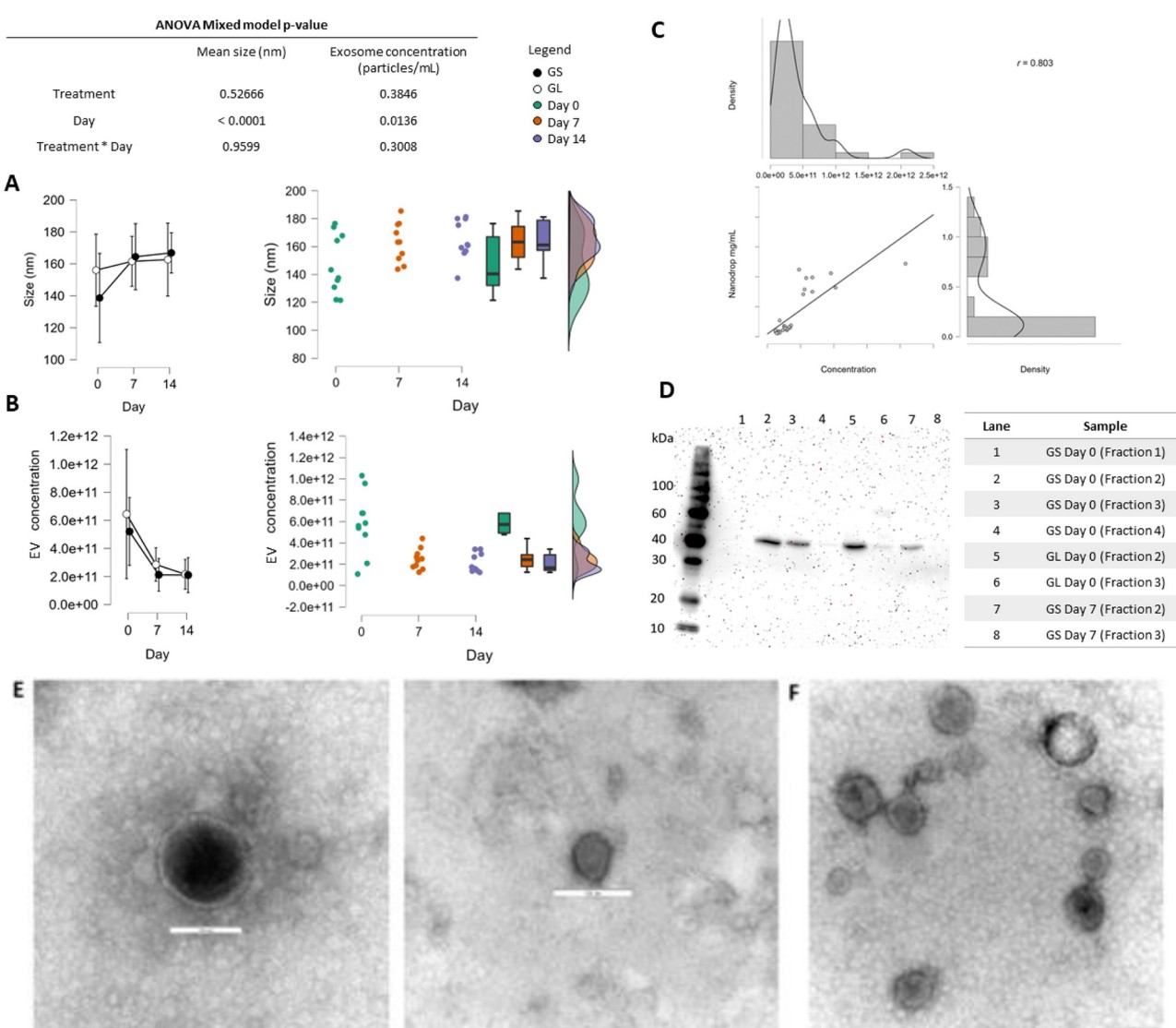

**Fig 1. Characterization of porcine milk extracellular vesicles (EV) in different stages of lactation from sows receiving diets with either a high (13:1, soybean oil [GS]) or a low (4.6:1, linseed oil [GL]) ratio of omega-6 to omega-3 (ω-6:ω-3) fatty acids.** A. The ANOVA mixed model comparison using data from nanoparticle tracking analysis (NTA) of mean size (nm) and EV concentration (particles/mL) in relation to the diets, time, and interaction of diet and time, with raincloud plots showing data distribution. Data are presented as means ± SEM. B. in relation to the diets, time, and interaction of diet and time, and raincloud plots with respective data distribution. Data are presented as means ± SEM. C. Pearson's correlation test ($R^2$ = 0.802, p-value < 0.05) comparing protein concentration (in mg/mL) and EV concentration (in particles/mL); D. Western blotting (WB) of EV marker TSG 101 in samples from both groups after ultracentrifugation coupled with size exclusion chromatography (SEC). Fractions 2 and 3 from SEC contained EV and were pooled together for further analysis. E. Representative electron micrograph of porcine milk exosomes in a close-up of a single vesicle and wide view. F. The bar scale is equal to 100 nm in all micrographs.

been previously identified by LC-MS/MS (Ferreira et al., 2021). The EV concentration was positively correlated with the protein concentration (Pearson's $R^2$ = 0.802, p-value < 0.05; Fig 1C) of the samples, corroborating a feasibly pure isolation free of contamination from other milk biocomponents. The examination by TEM showed round and cup-like concave spheres with morphology compatible with small EV. Representative images in wide and in close-up views are provided in Fig 1. Additional WB images are presented in S4 Fig.

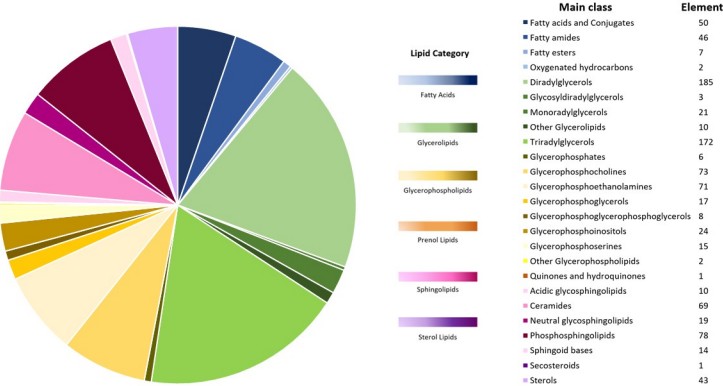

**Fig 2. Summary of 947 lipid elements identified in colostrum and milk extracellular vesicles (EV) by liquid chromatography–quadrupole time-of-flight mass spectrometry lipidomics analysis, color-coded by lipid category and stratified in lipid main classes.**

## Lipidomics analysis

The liquid chromatography–quadrupole time-of-flight mass spectrometry lipidomics analysis identified 947 lipids in milk and colostrum EV samples. The lipid elements identified covering six categories were divided into 25 main classes (Fig 2) and 47 subclasses. S2 Table summarizes the complete list of analyzed lipids within their respective lipid species classes, subclasses, and the number of lipids annotated from the untargeted lipidomic analysis. The most abundant lipid classes included: (1) diradylglycerols, a glyceride consisting of two fatty acid chains covalently bonded to a glycerol molecule through ester linkages; (2) triradylglycerols, formed by linking fatty acids with an ester linkage to three alcohol groups in glycerol; and (3) phosphosphingolipids, consisting of sphingolipids, mainly sphingomyelins, that include a phosphoryl group.

The principal component analysis indicated a clear separation between the time of sampling (day 0 vs. 7 and 14) and no separation between treatments (GS and GL) based on the lipid composition of EV from colostrum and milk (Fig 3).

The hierarchical clustering heat map of the top 50 significant differences at day 0, 7, and 14 when sows were fed diets with different ratios of ω-6 to ω-3 fatty acids showed a distinct lipidomic profile between colostrum and milk EV (Fig 4).

We observed 734 significantly different expressed lipid elements between day 0 and day 7 (722 down- and 12 up-regulated) in the lipid composition of EV from colostrum and milk (S3 Table). Compared with EV from milk at day 7, EV of colostrum had less DG (n = 176), TG (n = 70), PC (n = 62), SM (n = 50), FA (n = 48), and PE (n = 48) (Fig 5). In the contrast between day 0 and day 14, we observed 779 significantly different lipids (767 down- and 12 up-regulated; S4 Table).

Compared with EV from milk on day 14, EV of colostrum had lower DG (n = 193), TG (n = 81), PC (n = 61), Cer (n = 61), PE (n = 60), SM (n = 58), and FA (n = 48) (Fig 6). We observed no significant differences in the lipid composition of milk EV between days 7 and 14 (S2 Fig). Moreover, most of the up-regulated and down-regulated lipid elements in comparison between colostrum EV versus milk EV from day 7 were shared with the comparison between colostrum EV versus milk EV from day 14 (S3 Fig), evidencing distinct lipid signatures between colostrum and mature milk EV.

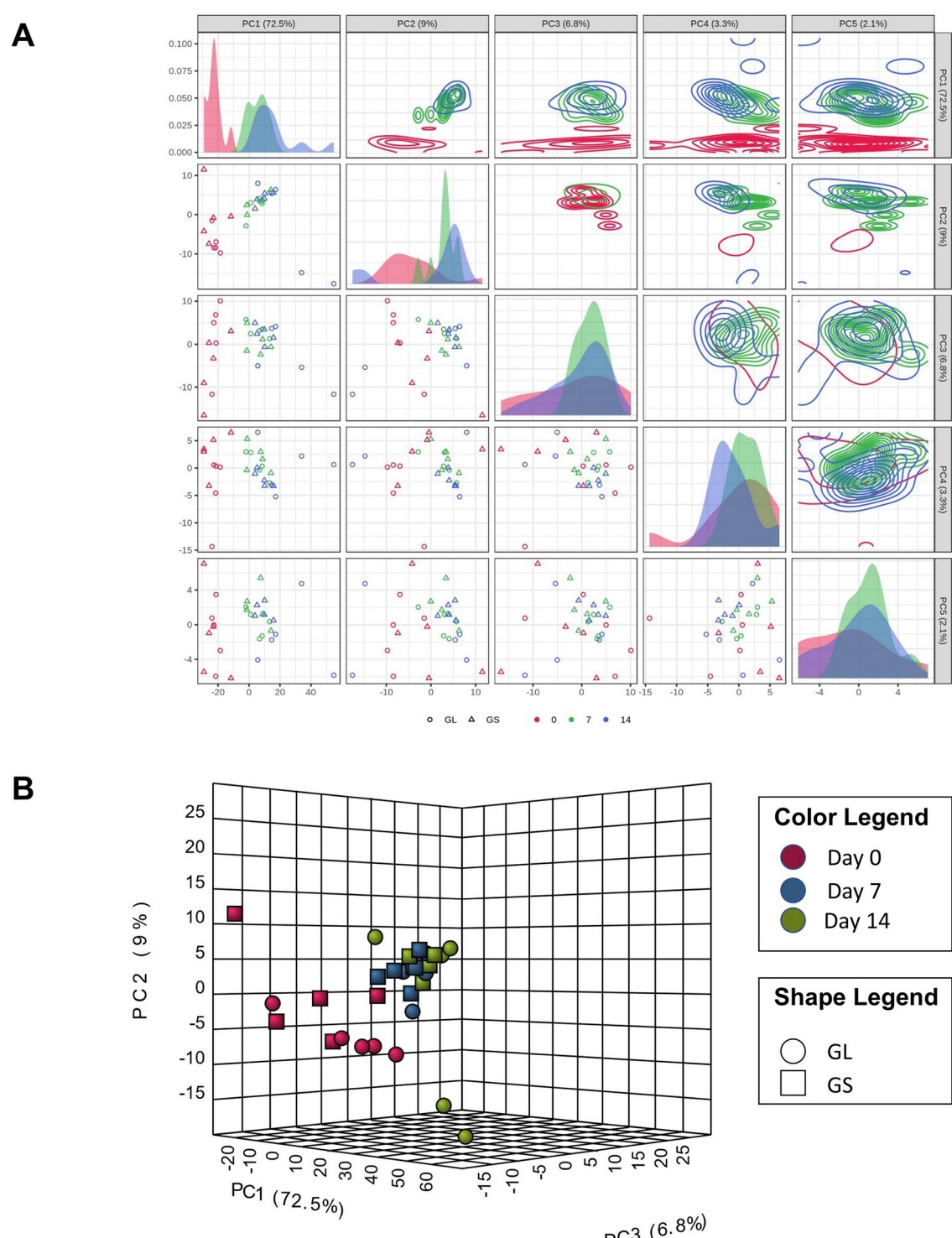

**Fig 3. Principal component analysis (PCA) plot illustrating the lipidomic profiling of extracellular vesicles (EV) from porcine colostrum and milk, showing a distinct lipidomic profile between colostrum (day 0) and mature milk (days 7 and 14) EV.** There was no clear separation between feeding groups receiving diets with either a high (13:1, soybean oil [GS]) or a low (4.6:1, linseed oil [GL]) ratio of omega-6 to omega-3 (ω-6:ω-3) fatty acids. The color legend differentiates the time points: red for day 0, blue for day 7, and green for day 14. The shape legend distinguishes the dietary groups: circles for the GL group and squares for the GS group.

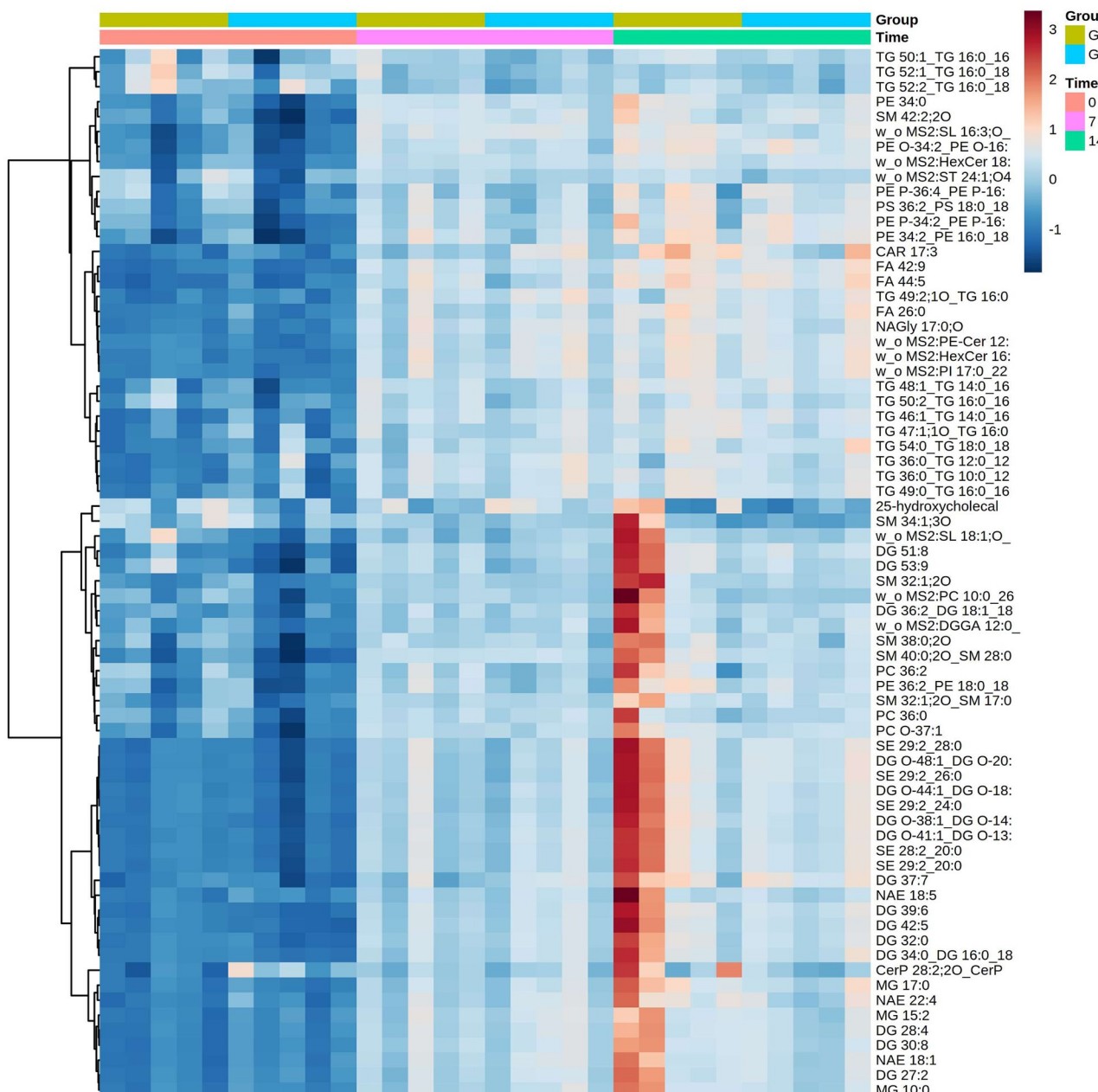

**Fig 4. Hierarchically clustered heatmap of the top 50 statistically significant lipids with the smallest P-values, grouped by day and time in milk extracellular vesicles (EV) from sows fed diets with either a high (13:1, soybean oil [GS]) or a low (4.6:1, linseed oil [GL]) ratio of omega-6 to omega-3 (ω-6:ω-3) fatty acids on days 0, 7, and 14.** The colors in the heat map reflect the different abundance of the lipids in milk EV and colostrum EV (mean smashed and divided by the range of each variable).

In addition, we examined differences in lipid composition between the treatments with different feeding ratios of ω-6 to ω-3 fatty acids. No differently abundant lipid element was identified in milk EV between treatments at day 0 and 7 (Fig 7A and 7B), and only one lipid (PA 17:0_28:6) was down-regulated in the comparison between milk EV from GS and GL at day 14 (Fig 7C).

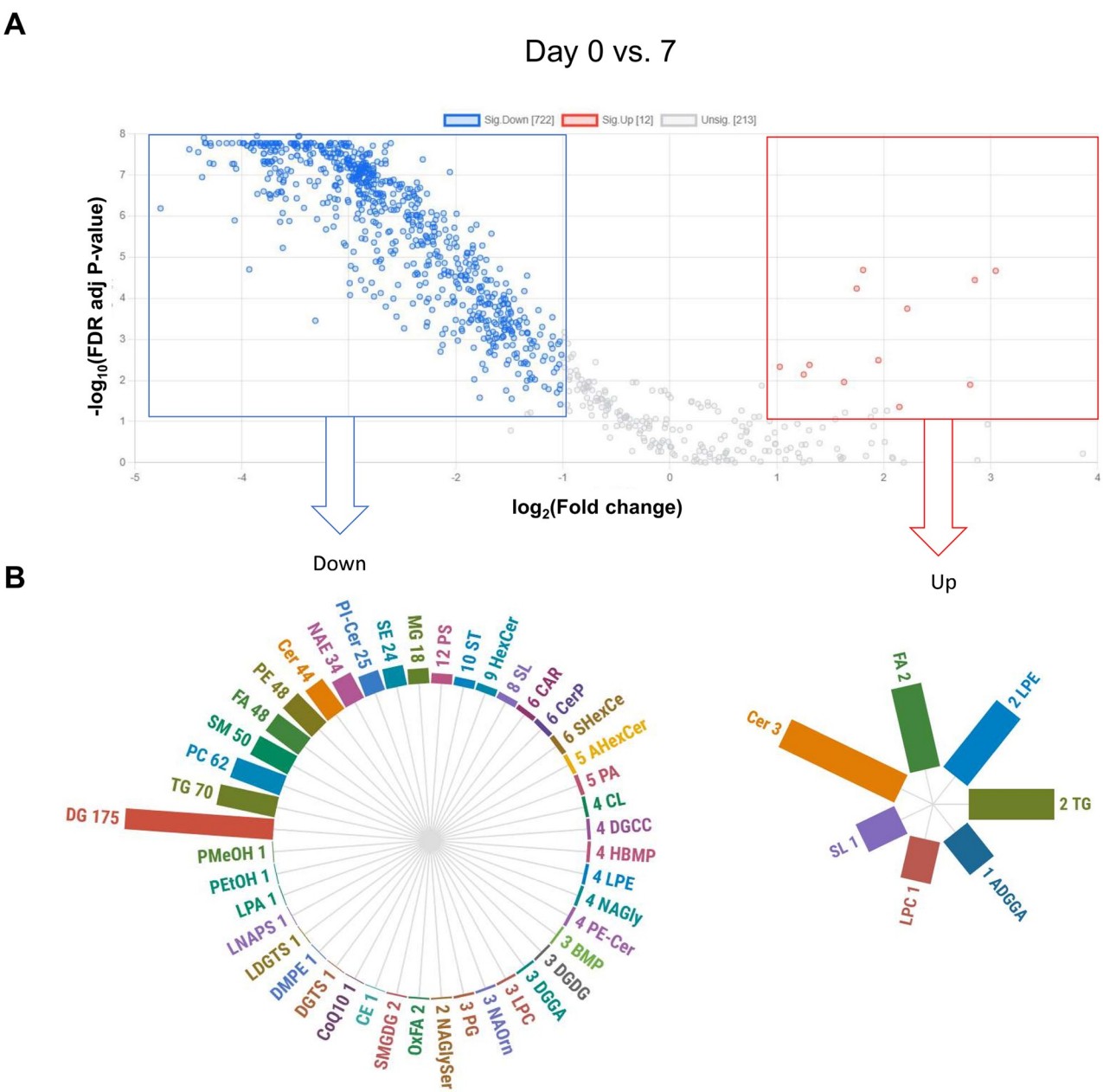

**Fig 5.** (A) Volcano plot of log$_2$ fold changes (x-axis) and their associated -log$_{10}$ false discovery rate (FDR) adjusted p-values (y-axis) of all identified lipids in the comparison between colostrum extracellular vesicles (EV) (day 0) and milk EV at day 7. Significantly different lipids (t-test FDR adjusted p-value < 0.05, fold change > 2) are highlighted, indicating up (red) and down (blue) regulated differentially expressed lipids. (B) The circular bar plots represent the number of lipid elements in each subclass that were up- or down-regulated on EV from colostrum compared to milk at day 7.

## Discussion

In this study, we investigated the effects of dietary ω-6:ω-3 ratio on the lipid composition of milk EV. We hypothesized that increasing the proportion of ω-3 fatty acids in the diet would alter the lipid profiles of milk EV, which could modify their structure and function. However, our results showed no significant differences in milk EV lipids between treatment diets. This suggests that milk EV lipids are more tightly regulated and more resilient to dietary changes

## Day 0 vs. 14

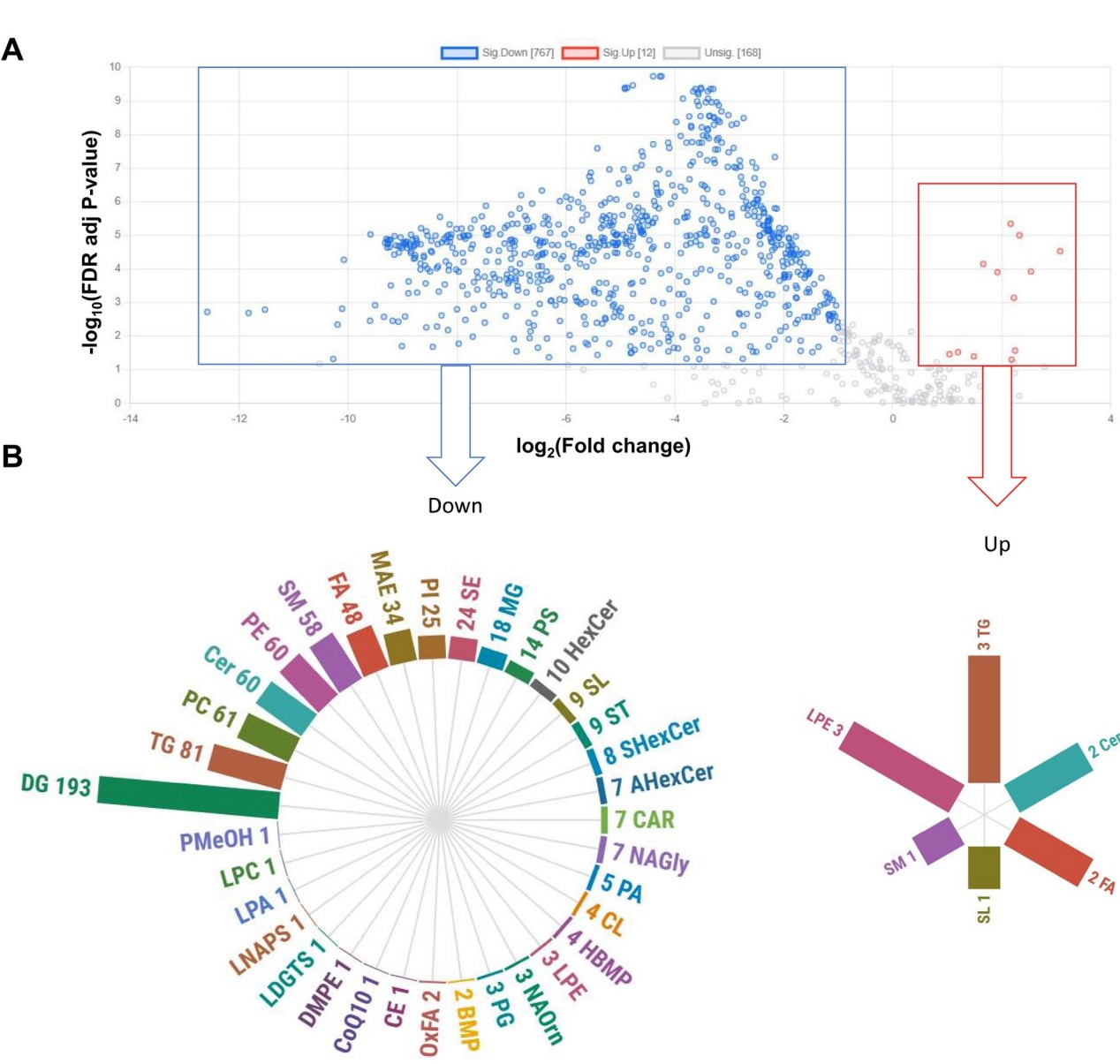

**Fig 6.** (A) Volcano plot of log$_2$ fold changes (x-axis) and their associated -log$_{10}$ false discovery rate (FDR) adjusted p-values (y-axis) of all identified lipids in the comparison between colostrum extracellular vesicles (EV) (day 0) and milk EV at day 14. Significantly different lipids (t-test FDR adjusted p-value < 0.05, fold change > 2) are highlighted, indicating up (red) and down (blue) regulated differentially expressed lipids. (B) The circular bar plots represent the number of lipid elements in each subclass that were up- or down-regulated on EV from colostrum compared to milk at day 14.

than originally hypothesized in this study. Consequently, diet composition can be based on factors such as cost or availability without compromising the quality of milk EV lipids.

Lipids play an important structural role in the EV membrane. Besides conveying essential nutrients do not present in the lipid MFG, they are also critical informational molecules crucial to fulfilling pivotal roles for EV formation, release, targeting, and uptake [37, 38, 69]. In comparison, EV have been characterized extensively in terms of their protein and nucleic acid

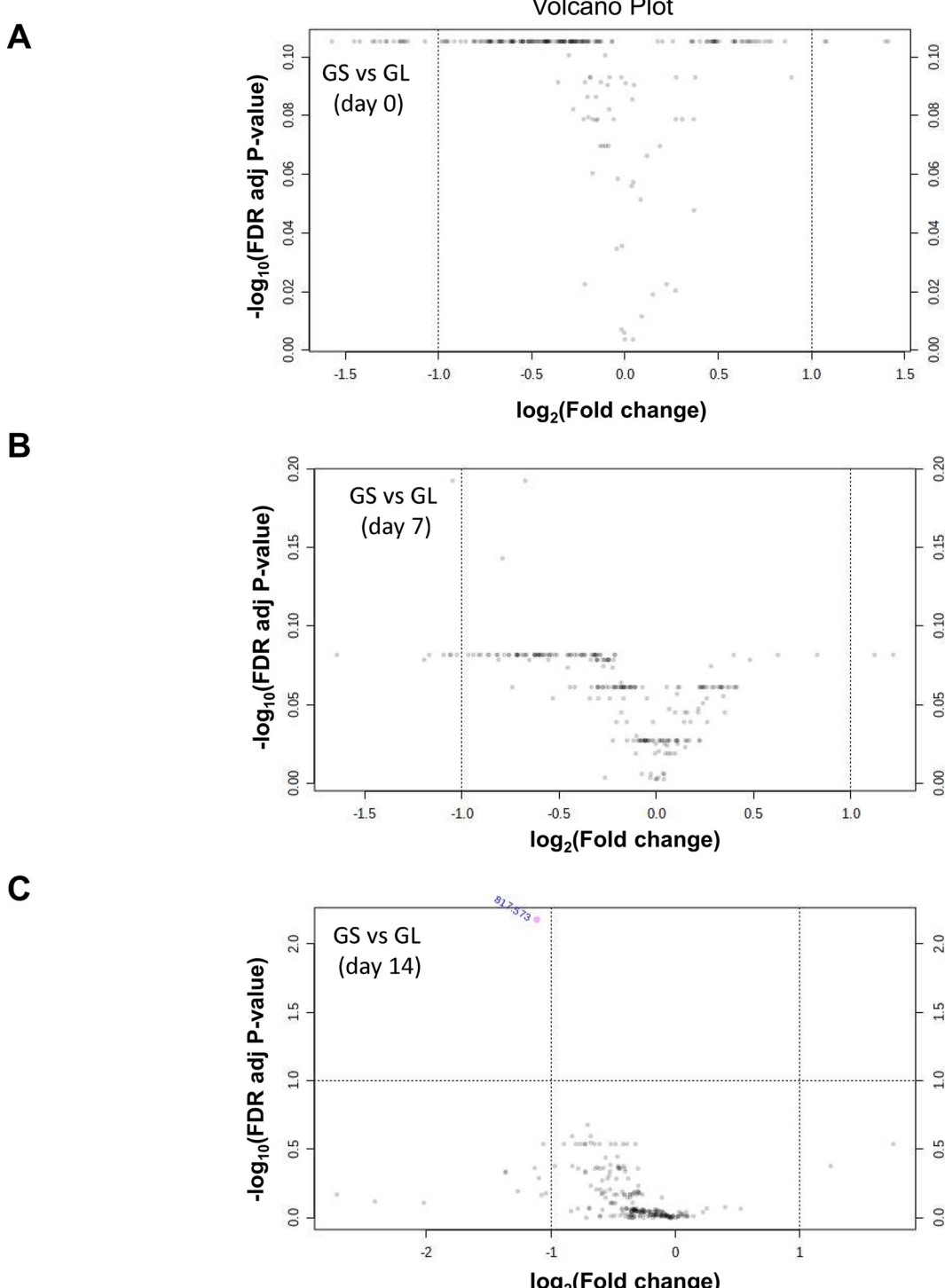

**Fig 7.** Volcano plots showing the log₂ fold changes (x-axis) and their associated -log₁₀ false discovery rate (FDR) adjusted p-values (y-axis) for all identified lipids in the comparison of milk extracellular vesicles (EV) from sows fed with soybean oil (GS) and linseed oil (GL) at different lactation stages: A. Day 0, B. Day 7, and C. Day 14 postpartum. No significant lipid differences were observed at time points A and B, while one lipid was found to be downregulated in the comparison between GS and GL at day 14 (C).

(mRNA and microRNA) content. However, their lipid components, including milk EV, have been largely overlooked [39]. The use of EV as drug carriers is highly sought, and while many attempts are being made, understanding their lipid composition and organization is a critical issue for their use as "natural liposomes" [70].

Nevertheless, there are some limitations to lipidomics studies with EV, mainly because it is not possible to produce completely pure EV isolates at the current state of the art. Sphingomyelins (SM), phosphatidylcholines (PC), ceramides (Cer), and cholesterol (Chol) are lipid classes that are abundant in cell membranes and, thus, also in EV. However, current knowledge of the lipid composition of EV from body fluids is partly empirical and based on knowledge of lipid organization in membranes and their subsequent physicochemical properties [71].

There are particular challenges in isolating lipids from milk-derived EV due to the inevitable co-isolation of milk lipids and MFG. Although the use of UV coupled with SEC has been used in milk EV research with positive results, contamination with free lipids is still considered unavoidable. The MFG contains many triacylglycerols (TG) in its core, potentially interfering with the interpretation of lipidomics results from milk EV [72]. TG were one of the significant subclasses identified in this study and one of the predominant subclasses of lipid elements that were differentially abundant when comparing milk EV and colostrum EV. To date, only one study [73] has examined the lipid composition of milk EV using an untargeted approach, in which TG was also found to be the most prominent lipid subclass. Studies have identified TG as a quantitatively insignificant intrinsic membrane component that plays a specific role in cellular stimulation and metastatic processes [62], and although it is plausible that some of the identified TG represent EV membrane-derived compounds, conclusions about EV composition should be drawn with caution because it is expected that many of these lipids are likely derived from membrane reminiscent of MFG. Lipid EV study results should be interpreted cautiously, highlighting lipids that fit the bilayer membrane structure formed by plasma or endosomal membranes [33].

SM, PC, Chol, and Cer represent the major lipid components of the EV membrane [63] and were identified in large numbers in our study. SM were the third class with the highest number of lipid elements identified here (69 lipid elements), with PC and Cer ranking fourth and fifth with 67 and 61 lipid elements, respectively. These classes were among the most significantly differentially abundant lipids between EV from colostrum and mature milk at 7 and 14. These results suggest that porcine colostral EV and milk EV have different lipid profiles and that exosomal membrane composition changes during the lactation phases.

SM are an important class of phospholipids identified in the membrane of EV of all cell types [40]. The EV membrane is also enriched in SM compared to their parent cells [71]. However, EV lipid content and specific lipid enrichment depend on the cell of origin. SM was found to be more enriched in B lymphocytes and dendritic cells than in oligodendroglial precursor cells [38]. Mammary gland tissue is particularly rich in SM, both in the lactating and non-lactating phases [74], and is, therefore, expected to be enriched in milk EV. However, no data in the literature compares it to EV in distinct body fluids. The SM provide EV stability and structural rigidity [37] but are also associated with EV function. It has been reported that the angiogenic activity of tumor-derived EV is mainly mediated by SM both *in vitro* and *in vivo* [75]. The underlying effects of the down-regulation of SM found in colostrum-derived EV require further investigation.

In the current study, we observed that EV increased in size during the transition from colostrum to mature milk whereas their concentration decreased with significant changes in lipid composition, especially an increased PC content. The larger EV size in mature milk also results in a larger membrane surface area, but the concomitantly decreasing EV particle

concentration would result in decreased membrane concentrations in the EV preparations. This reflects a relatively greater PC concentration in the membranes from EV in mature milk than from colostrum. PC are critical to the structure and stability of the EV [39], facilitating their binding to recipient cells and improving cellular uptake [76]. The overproportional increase in PC observed herein may thus contribute to greater membrane fluidity which in turn enables more effective fusion with cell membranes and promotes efficient uptake and rapid release of bioactive components via direct fusion pathways [77]. Studies by Zhang et al [78] have shown that lipid diversity in exosomes varies by size and depends on the cell type of origin, with PC accounting for 46%–89% of lipid components in all EV, and that higher PC content are associated with greater utilization of direct fusion mechanisms. These changes in lipid composition in EV may be associated with early life programming in neonates, emphasizing the significance of lipid modifications in EV for enhancing neonatal development.

Cer is one of the essential lipids found in intraluminal vesicles (ILV) and on the multivesicular endosomes (MVE) membrane, as it is involved in the inward budding of endosomes to form multivesicular bodies [79, 80]. As a result, the inner leaflet of the EV membrane is enriched in Cer [81]. Many Cer elements were down-regulated in colostrum EV compared with milk EV (53 Cer at day 7 and 60 at day 14). The structure of the lipid bilayer has a direct influence on its rigidity and, thus, on the size of the EV. Colostrum EV was significantly smaller than milk EV, as demonstrated in two different isolation batches, and modulation of the lipid components SM, PC, and Cer may be responsible for this effect.

Chol is one of the major structural components of the plasma membrane of cells and, thus, of EV [82, 83]. Although it was identified in all EV samples in this study, its abundance did not differ between lactation stages. A substantial number of other sterol components (39 lipid elements) were identified in both milk and colostrum EV. Sterols are considered membrane reinforcers because they maintain the domain structure of the cell membrane [84]. A total of 7 differentially abundant sterols were identified to be down-regulated in both colostrum EV and milk EV at days 7 and 14, namely stigmasterols (ST 29:2;O;Hex; FA 20:1, ST 29:1;O;Hex; FA 15:2, and ST 29:1;O;Hex; FA 13:0) and steroid conjugates (ST 24:1;O4_19:2;1O, ST 24:2; O4_2:0, ST 24:1;O4;G_16:2;1O, ST 24:1;O3;G_28). Stigmasterols are significant components of the sterol profiles of plant species [85], and although cholesterol is the predominant sterol in animal membranes, plant sterols (phytosterols) can also be found [86]. Phytosterols are absorbed from the diet and incorporated into cell membranes. However, because their uptake through the intestinal tract is very low, their content in mammalian membranes is relatively low [87, 88].

Sterols and sphingolipids are also crucial for forming lipid rafts, which play an important role in signal transduction, cytoskeleton reorganization, and cell sorting [89]. In particular, in EV, lipid rafts are involved in both cargo loading via an endosomal sorting complex required for transport machinery (ESCRT)-the independent mechanism and the shift of proteins and molecules that facilitate their secretion [69, 89]. Thus, they are directly involved in EV function in intercellular communication. Many lipid compounds down-regulated in colostrum EV were involved in lipid raft composition, which could directly affect EV biogenesis and EV trafficking.

## Conclusions

The lipidomic characterization provided insights into the structure, function, and stability properties of porcine milk EV. The major lipid components of the EV membrane, such as SM, PC, Chol, and Cer, were identified in our results and were downregulated in colostrum EV compared with milk EV. Lipid elements associated with lipid rafts were also among the

significant lipids identified in this study. These lipids are involved in numerous exosome biogenesis and secretion processes, including a cargo sorting mechanism that is not dependent on the ESCRT machinery. This study helps to fill the gap in the limited knowledge of lipid characterization of milk EV, and the underlying effects of lipid modulation on EV structure and function remain unexplored. The complete lipidome characterization of milk EV at different stages of lactation presented here is also of great importance for innovative bionanotechnological approaches, as milk EV are proposed as a potential natural drug delivery system.

## Supporting information

**S1 Fig. Representative ion chromatograms from QC samples in positive (S1A, S1B Fig) and negative (S1C, S1D Fig) ionization modes to ensure data quality and consistency across the lipidomic analysis.** https://doi.org/10.6084/m9.figshare.28016336.v2.
(PDF)

**S2 Fig. Volcano plot illustrating $\log_2$ fold changes and $-\log_{10}$ FDR-adjusted p-values for lipids in milk EV from days 7 and 14 postpartum, showing no significant differences in lipid composition between these time points.** https://doi.org/10.6084/m9.figshare.28016378.v1.
(PDF)

**S3 Fig. Venn diagram comparing differentially expressed lipids in colostrum EV (day 0) versus milk EV at days 7 and 14, highlighting shared and unique lipid elements.** https://doi.org/10.6084/m9.figshare.28016396.v1.
(PDF)

**S4 Fig. Western blot analysis of the EV marker TSG101 to confirm successful EV isolation in both dietary groups after ultracentrifugation and size exclusion chromatography.** https://doi.org/10.6084/m9.figshare.28016408.v1.
(PDF)

**S1 Table. Composition of basal sow diets, including ingredient breakdown, nutrient composition, and fatty acid profiles, detailing differences in ω-6:ω-3 ratios.** https://doi.org/10.6084/m9.figshare.28016372.v2.
(PDF)

**S2 Table. Classification of lipids by category, main class, and subclass, including proportions of each lipid class identified in the untargeted lipidomic analysis.** https://doi.org/10.6084/m9.figshare.28016393.v1.
(PDF)

**S3 Table. Summary of significantly up- and down-regulated lipids in porcine colostrum EV (day 0) versus milk EV at day 7, with detailed fold changes and FDR-adjusted p-values.** https://doi.org/10.6084/m9.figshare.28016402.v1.
(PDF)

**S4 Table. Significantly up- and down-regulated lipids in colostrum EV (day 0) versus milk EV at day 14, providing lipid names, fold changes, and FDR-adjusted p-values.** https://doi.org/10.6084/m9.figshare.28016414.v1.
(PDF)

## Author Contributions

**Conceptualization:** Rafaela Furioso Ferreira, Fabrizio Ceciliani, Donatella Caruso, Vladimir Mrljak, Helga Sauerwein.

**Data curation:** Rafaela Furioso Ferreira, Morteza H. Ghaffari, Manuela Fontana, Matteo Audano.

**Formal analysis:** Rafaela Furioso Ferreira, Morteza H. Ghaffari, Manuela Fontana, Donatella Caruso.

**Funding acquisition:** Vladimir Mrljak, Helga Sauerwein.

**Investigation:** Rafaela Furioso Ferreira, Fabrizio Ceciliani, Donatella Caruso, Matteo Audano, Giovanni Savoini, Alessandro Agazzi.

**Methodology:** Rafaela Furioso Ferreira, Manuela Fontana, Donatella Caruso, Matteo Audano, Giovanni Savoini, Alessandro Agazzi.

**Project administration:** Rafaela Furioso Ferreira, Helga Sauerwein.

**Resources:** Giovanni Savoini, Alessandro Agazzi, Vladimir Mrljak.

**Supervision:** Vladimir Mrljak, Helga Sauerwein.

**Writing – original draft:** Rafaela Furioso Ferreira, Helga Sauerwein.

**Writing – review & editing:** Morteza H. Ghaffari, Fabrizio Ceciliani, Donatella Caruso, Giovanni Savoini, Alessandro Agazzi, Vladimir Mrljak.

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
