## [Decision Letter · Decision Letter 0]

5 Aug 2024

PONE-D-24-15462Untargeted lipidomics reveals unique lipid signatures of extracellular vesicles from porcine colostrum and milkPLOS ONE

Dear Dr. Furioso Ferreira,

Thank you for submitting your manuscript to PLOS ONE. After careful consideration, we feel that it has merit but does not fully meet PLOS ONE’s publication criteria as it currently stands. Therefore, we invite you to submit a revised version of the manuscript that addresses the points raised during the review process.

We look forward to receiving your revised manuscript.

Kind regards,

Alok Raghav, PhD

Academic Editor

PLOS ONE

Journal Requirements:

https://doi.org/10.1016/j.jprot.2022.104632

In your revision ensure you cite all your sources (including your own works), and quote or rephrase any duplicated text outside the methods section. Further consideration is dependent on these concerns being addressed.

"EXCELLENT SCIENCE - Marie Skłodowska-Curie Actions - Grant agreement ID: 765423"

**Additional Editor Comments:**

The authors should address the reviewers comments

Reviewers' comments:

Reviewer's Responses to Questions

**Comments to the Author**

1. Is the manuscript technically sound, and do the data support the conclusions?

Reviewer #1: Yes

Reviewer #2: Partly

2. Has the statistical analysis been performed appropriately and rigorously? 

Reviewer #1: Yes

Reviewer #2: Yes

3. Have the authors made all data underlying the findings in their manuscript fully available?

Reviewer #1: Yes

Reviewer #2: No

4. Is the manuscript presented in an intelligible fashion and written in standard English?

Reviewer #1: Yes

Reviewer #2: Yes

5. Review Comments to the Author

Reviewer #1: Reviewer’s comments:

Using the swine model, the manuscript “PONE-D-24-15462” currently under review showed that although the lipid composition of milk extracellular vesicles (EV) was not affected by the maternal diet (i.e., different ratios of ω-6:ω-3 fatty acids in the gestation and lactation diets of sows), distinct extracellular vesicle lipidomic profile were found in colostrum compared to mature milk collected at 7 and 14 days after parturition.

The authors used ultracentrifugation coupled with size exclusion chromatography to isolate EV, nanoparticle tracking analysis and transmission electron microscopy to characterize EV, and assessment of EV markers via Western blotting to assess the EV markers, and liquid chromatography–quadrupole time-of-flight mass spectrometry approach to determine the lipodome.

While it is currently not possible to produce completely pure EV isolates, the manuscript is an important development research to provide a better understanding of the use of porcine milk EV as drug carriers (as "natural liposomes") in a potential natural drug delivery system.

The manuscript was generally well-written although the number of tables and figures may be reduced. Some revisions are suggested as follows:

L48-49, write: … multiple cellular sources (12), MFG are ... Instead of multiple cellular sources,

(12) MFG are …

L121, write: … (48) ... Instead of … (47),

L288, write: … main classes (Figure 2) … Instead of … main classes (Fig 2) …

L304, write: … (Supplemental Table S1) … Instead of … (Supplemental Table S4) …

L307, write: … (Supplemental Table S2) … Instead of … (Supplemental Table S5) …

L325, write: … and uptake (37, 38, 72). Instead of … and uptake (37, 38; 72).

L343, please insert reference number in parenthesis Instead of … only one study () has examined …

L357, write: … have different lipid profiles ... Instead of … have different profiles ...

L369-370, insert paragraph to discuss/explain results related to PC (phosphatidylcholine)

L454-679, please check/correct all entries under References following standard journal format and font type (L481-490, L585-599, and L607-608) required by PLOS One.

L701-704, please indicate what A and B charts are all about.

Reviewer #2: - The time-point comparisons are not the same: 0 vs 7 and 0 vs 14 use 1.5 lfc (Fig 6, 7), while 7 vs 14 use 2 lfc (Supp fig 1), please recheck the criteria

- Although GS vs. GL has a negative result, the principles of why two different treatments were included in this study and how they should hypothetically affect the lipids in the milk should be discussed. This could at least inform that the different feeding recommendations (might be GL better than GS at first, etc.) did not change the milk composition and can be chosen based on the cost perspective.

- Since UC and SEC are used, this means that the contamination of free lipids is unavoidable. This statement should be declared in the limitation

- There was a missing citation at line 343.

- Please provide the following data for the reproducible analysis

o Data processing parameters in MS-DIAL, e.g. MS1/MS2 tolerance, identification cut-off, adduct used

o Raw files (on the repository, e.g. https://www.metabolomicsworkbench.org/) and their corresponding search result for the reproducible analysis

- Please provide abbreviations in the captions

6. PLOS authors have the option to publish the peer review history of their article (what does this mean?). If published, this will include your full peer review and any attached files.

Reviewer #1: **Yes: **Orville L. Bondoc

Reviewer #2: No

---

## [Author Response · Author response to Decision Letter 0]

28 Oct 2024

Dear Editor,

Thank you for your feedback and the opportunity to revise our manuscript. We appreciate the reviewers' insightful comments and have addressed each point thoroughly. We have provided a detailed response letter addressing each reviewer comment point by point, along with the revised manuscript highlighting all changes made.

Reviewer's Responses to Questions

Comments to the Author

1. Is the manuscript technically sound, and do the data support the conclusions?

Reviewer #1: Yes

Reviewer #2: Partly

AU: We appreciate Reviewer #1's positive feedback. In response to Reviewer #2's concerns, we have revised and strengthened the discussion of our findings to better support the conclusions drawn from the data.

2. Has the statistical analysis been performed appropriately and rigorously?

Reviewer #1: Yes

Reviewer #2: Yes

AU: Thank you for confirming the appropriateness of our statistical analysis.

3. Have the authors made all data underlying the findings in their manuscript fully available?

Reviewer #1: Yes

Reviewer #2: No

AU: We have addressed Reviewer #2's concern by providing a detailed Data Availability Statement. We have deposited the raw data in a public repository and included all necessary data processing parameters in the supplementary materials to ensure reproducibility. The raw data from positive and negative ion modes are reported in https://doi.org/10.13130/RD_UNIMI/UZI5O8 and https://doi.org/10.13130/RD_UNIMI/0HA5IL respectively.

4. Is the manuscript presented in an intelligible fashion and written in standard English?

Reviewer #1: Yes

Reviewer #2: Yes

AU: We appreciate the positive feedback on clarity and have made minor edits to enhance readability.________________________________________

5. Review Comments to the Author

Response to Reviewer #1’s comments:

Using the swine model, the manuscript “PONE-D-24-15462” currently under review showed that although the lipid composition of milk extracellular vesicles (EV) was not affected by the maternal diet (i.e., different ratios of ω-6:ω-3 fatty acids in the gestation and lactation diets of sows), distinct extracellular vesicle lipidomic profile were found in colostrum compared to mature milk collected at 7 and 14 days after parturition. The authors used ultracentrifugation coupled with size exclusion chromatography to isolate EV, nanoparticle tracking analysis and transmission electron microscopy to characterize EV, and assessment of EV markers via Western blotting to assess the EV markers, and liquid chromatography–quadrupole time-of-flight mass spectrometry approach to determine the lipodome. While it is currently not possible to produce completely pure EV isolates, the manuscript is important development research to provide a better understanding of the use of porcine milk EV as drug carriers (as "natural liposomes") in a potential natural drug delivery system.

AU: We sincerely thank the reviewer for their thorough evaluation of our manuscript and for the valuable feedback provided. We have revised address the points raised and have highlighted major changes in yellow within the revised manuscript. Detailed responses to the comments are provided below, prefaced with "AU" for "authors." Line numbers reflect the “Simple markup” function in Word.

The manuscript was generally well-written although the number of tables and figures may be reduced. Some revisions are suggested as follows:

AU: We have revised the figures by removing the previous Figure 3A (a portion of the PCA plot) and relocating the former Figure 7 to Supplemental Figure S1. This adjustment decreases the total number of figures while retaining all essential information.

L48-49, write: … multiple cellular sources (12), MFG are ... Instead of multiple cellular sources, (12) MFG are … L121, write: … (48) ... Instead of … (47), L288, write: … main classes (Figure 2) … Instead of … main classes (Fig 2) … - L304, write: … (Supplemental Table S1) … Instead of … (Supplemental Table S4) …L307, write: … (Supplemental Table S2) … Instead of … (Supplemental Table S5) …L325, write: … and uptake (37, 38, 72). Instead of … and uptake (37, 38; 72). L343, please insert reference number in parenthesis Instead of … only one study () has examined …

L357, write: … have different lipid profiles ... Instead of … have different profiles ...

AU: All suggested corrections (e.g., L48-49, L121, L288, etc.) have been implemented. The manuscript has been revised to reflect these changes.

L369-370, insert paragraph to discuss/explain results related to PC (phosphatidylcholine)

AU: Thank you for your suggestion. We have added the following paragraph to discuss the results related to phosphatidylcholine: “In the current study, we observed that EV increased in size during the transition from colostrum to mature milk whereas their concentration decreased with significant changes in lipid composition, especially an increased PC content. The larger EV size in mature milk also results in a larger membrane surface area, but the concomitantly decreasing EV particle concentration would result in decreased membrane concentrations in the EV preparations. This reflects a relatively greater PC concentration in the membranes from EV in mature milk than from colostrum. PC are critical to the structure and stability of the EV [39], facilitating their binding to recipient cells and improving cellular uptake [76]. The overproportional increase in PC observed herein may thus contribute to greater membrane fluidity which in turn enables more effective fusion with cell membranes and promotes efficient uptake and rapid release of bioactive components via direct fusion pathways [77]. Studies by Zhang et al. [78] have shown that lipid diversity in exosomes varies by size and depends on the cell type of origin, with PC accounting for 46%–89% of lipid components in all EV, and that higher PC content are associated with greater utilization of direct fusion mechanisms. These changes in lipid composition in EV may be associated with early life programming in neonates, emphasizing the significance of lipid modifications in EV for enhancing neonatal development.” Lines 373-388

L454-679, please check/correct all entries under References following standard journal format and font type (L481-490, L585-599, and L607-608) required by PLOS One.

AU: Thank you for your comment. All references have been reviewed and corrected to adhere to the standard format and font type required by PLOS ONE.

L701-704, please indicate what A and B charts are all about.

AU: We have removed Panel A to reduce the number of figures as requested but we have added descriptions to clarify the content of the PCA plot. 

Reviewer #2: - The time-point comparisons are not the same: 0 vs 7 and 0 vs 14 use 1.5 lfc (Fig 6, 7), while 7 vs 14 use 2 lfc (Supp fig 1), please recheck the criteria

AU: Thank you for your comment on the fold change thresholds. For all comparisons presented in Supplemental Figure S1 we have consistently used a fold change of 2 (log2FC) = 1). The horizontal lines at 1 and -1 on the log2FC axis in the graphs indicate the threshold for significance. We thank you for your attention to this detail and have verified that these criteria have been applied consistently to all comparisons.

- Although GS vs. GL has a negative result, the principles of why two different treatments were included in this study and how they should hypothetically affect the lipids in the milk should be discussed. This could at least inform that the different feeding recommendations (might be GL better than GS at first, etc.) did not change the milk composition and can be chosen based on the cost perspective.

AU: Thank you for your valuable feedback. We have incorporated the following hypothesis in lines 84-87: “This study provides new insights into the lipidomic characterization of porcine milk EV, filling a significant gap in our understanding of how maternal nutrition influences the lipid composition of these vesicles. We hypothesized that the intake of two different diets, grain silage and grass legumes, characterized by a different ratio of ω-6:ω-3 fatty acids, would significantly alter the lipid composition of milk vesicles”. Additionally, in the discussion section (lines 317-323), we state: “In this study, we investigated the effects of dietary ω-6:ω-3 ratio on the lipid composition of milk EV. We hypothesized that increasing the proportion of ω-3 fatty acids in the diet would alter the lipid profiles of milk EV, which could modify their structure and function. However, our results showed no significant differences in milk EV lipids between treatment diets. This suggests that milk EV lipids are more tightly regulated and more resilient to dietary changes than originally hypothesized in this study. Consequently, diet composition can be based on factors such as cost or availability without compromising the quality of milk EV lipids.”

- Since UC and SEC are used, this means that the contamination of free lipids is unavoidable. This statement should be declared in the limitation

AU: Thank you for your comment. We agree with that limitation of the study, which has been partially addressed in lines 333-334: “Nevertheless, there are some limitations to lipidomics studies with EV, mainly because it is not possible to produce completely pure EV isolates at the current state of the art.” And lines 340-341: “There are particular challenges in isolating lipids from milk-derived EV due to the inevitable co-isolation of milk lipids and MFG”. We have prioritized to obtain a reliable level of purity in our EV isolations, by implementing a combination of two isolation techniques (ultracentrifugation coupled with size exclusion chromatography), which has been used in milk EV research with positive results. We agree, however, that the contamination of free lipids is yet unavoidable with current techniques, including the ones using our study, and therefore have included the following sentence in lines 341-343 in the limitations to make it clearer: “Although the use of UV coupled with SEC has been used in milk EV research with positive results, the contamination of free lipids is still considered unavoidable.” 

- There was a missing citation at line 343.

AU: Citation was added.

- Please provide the following data for the reproducible analysis

o Data processing parameters in MS-DIAL, e.g. MS1/MS2 tolerance, identification cut-off, adduct used

o Raw files (on the repository, e.g. https://www.metabolomicsworkbench.org/) and their corresponding search result for the reproducible analysis

AU: Thank you for your request for additional information regarding data reproducibility. We have included the following data processing parameters used in MS-DIAL. The raw data from positive and negative ion modes are reported in https://doi.org/10.13130/RD_UNIMI/UZI5O8 and https://doi.org/10.13130/RD_UNIMI/0HA5IL respectively. We appreciate your suggestion and are committed to ensuring our data can be reproduced effectively.

- Please provide abbreviations in the captions

AU: We have revised the figure legends to include all relevant abbreviations for clarity.

---

## [Editor Report · Decision Letter 1]

30 Oct 2024

Untargeted lipidomics reveals unique lipid signatures of extracellular vesicles from porcine colostrum and milk

PONE-D-24-15462R1

Dear Dr. Ferreira,

We’re pleased to inform you that your manuscript has been judged scientifically suitable for publication and will be formally accepted for publication once it meets all outstanding technical requirements.

Kind regards,

Alok Raghav, PhD

Academic Editor

PLOS ONE

---

## [Editor Report · Acceptance letter]

4 Jan 2025

PONE-D-24-15462R1 

PLOS ONE

Dear Dr. Furioso Ferreira, 

I'm pleased to inform you that your manuscript has been deemed suitable for publication in PLOS ONE. Congratulations! Your manuscript is now being handed over to our production team.

Kind regards, 

on behalf of

Dr. Alok Raghav 

Academic Editor

PLOS ONE